# Time to phlebitis onsite and its predictors among admitted patients with peripheral intravenous cannulas at Debre Markos Comprehensive Specialized Hospital, Northwest Ethiopia, 2024/2025: A prospective follow-up study

Zinaw Beyene[1☉], Haymanot Zeleke Mitiku[2☉*], Tesfahun Ayenew[2‡], Mengistu Abebe[2‡], Mihret Kefie[1‡], Afework Edmealem[2‡], Hailie Amha[2☉]

1 Debre Markos Comprehensive Specialized Hospital, Debre Markos, Amhara, Ethiopia, 2 Department of Nursing, College of Health Sciences, Debre Markos University, Debre Markos, Amhara, Ethiopia

☉ These authors contributed equally to this work.
‡ These authors also contributed equally to this work.
* haymanotzeleke89@gmail.com, haymanot_zeleke89@dmu.ed.et

## Abstract

### Introduction

Phlebitis due to peripheral intravenous cannulation is a common issue in hospitalized patients which leading to serious complications such as deep vein and pulmonary thromboembolism, and septicemia. Understanding the time to phlebitis onset and its predictors is of clinical importance. However, data on its incidence, contributing factors, and time of onset remain limited in Ethiopia. Given the scarcity of compressive studies conducted in Ethiopia and the critical knowledge gap on the incidence, predictors, and timing of phlebitis onset, this study is significant in advancing clinical practice by identifying the time to onset of phlebitis and its predictors among hospitalized patients with peripheral intravenous cannulation.

### Method

A prospective follow-up study design was conducted on a sample size of 372 participants using structured interviews and observational checklists. Time to phlebitis onset was determined with the Kaplan-Meier method and log-log plots, while predictors were identified using Cox proportional hazards regression.

### Result

The cumulative incidence of phlebitis was 39.25% (95% CI: 34.25%–44.41%), with a median onset time of 5 days (IQR 4–6 days). Whereas, the significant predictors were female sex (AHR = 1.58, 95% CI: 1.11–2.25), use of an 18-gauge cannula

**Data availability statement:** All relevant data are within the manuscript.

**Funding:** The author(s) received no specific funding for this work.

**Competing interests:** The authors have declared that no competing interests exist.

(AHR = 2.02, 95% CI: 1.20–3.41), receipt of a blood transfusion (AHR = 2.11, 95% CI: 1.14–3.91), and administration of potassium chloride (AHR = 1.93, 95% CI: 1.17–3.19) and vancomycin (AHR = 2.89, 95% CI: 1.73–4.83).

## Conclusion

In this study, the incidence of peripheral intravenous cannula-induced phlebitis was high, with a lower median time of phlebitis onset. The key predictors were female sex, large cannula, receipt of a blood transfusion, and administration of potassium chloride and vancomycin.

## Introduction

Peripheral intravenous cannulation is one of the most common invasive procedures performed in clinical setting [1]. Each year, over one billion intravenous cannulas are placed in patients worldwide [2]. While this procedure is essential for giving medications, fluid, blood transfusions, and supporting treatments and diagnostic procedures like hemodialysis, parenteral nutrition, and contrast studies, it can also cause complications, the most frequent of which is phlebitis [3].

Phlebitis is the inflammation of a vein and often occurs at the cannula insertion site. It may start with tenderness, erythema, swelling and can progress along the vein to form a palpable venous cord, purulent discharge, intense redness, and fever [4–6].

Several factors can contribute for occurrence of phlebitis. These are mechanical, chemical, and bacterial causes [7,8]. Mechanical factors, such as cannula size and insertion sit, can irritate the vein wall [8,9] and trigger inflammation [10,11]. Chemical factors also contribute, particularly the type of solution and the infusion speed, with the faster rates increasing the likelihood of vein irritation [12]. In addition, bacterial factors play a critical role [13], as improper sterile technique during cannula insertion can enter bacteria into the bloodstream, resulting to infection [13–15].

The exact global burden of phlebitis associated with peripheral intravenous cannula (PIVC) remains unclear and needs further investigation. However, studies report that its incidence ranges from 3% to 70%, with the highest rates observed in developing countries [16–19]. Findings from various countries show considerable variation in PIVC-related phlebitis. For instance, a study in Japan reported an incidence rate of 9.1% [18], other studies reported rates of 6.1% in Brazil, [20] 31% in a Turkish tertiary care center [21], 31.4% in India [9], 39.1% in Pakistan [22], and as high as 70% at Gonder hospital in Ethiopia [16].

Phlebitis has significant clinical and economic impacts on hospitalized patients in various ways. It can expose for unnecessary diagnostic procedures and treatments, increasing healthcare costs by an estimated $437 million annually and prolonging hospital stays [23]. A cohort study in China involving 1,069 hospitalized adults found that 10.5% of patients incurred additional medical costs for catheter replacement and management due to phlebitis [24]. Similarly, research on the clinical and economic burden of PIVC-related complications revealed that patients with complications

experienced hospital stays 51% longer than those without (5.9 days vs. 3.9 days) and incurred 55% higher hospitalization costs ($10,895 vs. $7,009) [25]. Beyond economic consequences, phlebitis can result in local and systemic infections, leading to serious complications such as pulmonary embolism, deep vein thrombosis, septicemia, and increased patient morbidity and mortality [19]. The same study reported that the mortality was five times higher in patients with complications comparted to those without (3.6% vs. 0.7%) [25].

It is recommended that the cubital fossa not be the first choice for cannulation when administering therapy and that the cannula insertion site be changed every 72 hours to minimize the risk of phlebitis. Additionally, strict adherence to aseptic techniques during cannula insertion and close monitoring for early indicators of phlebitis, such as changes in skin color, texture, and temperature, are strongly advised [14,26]. Despite these recommendations, phlebitis remains a common clinical issue. Moreover, there is a shortage of compressive studies conducted in this area, particularly in Ethiopia. Only a limited number of studies have assessed the incidence and predictors of phlebitis, with reported rates varying considerably [16,27]. Furthermore, there is a critical knowledge gap regarding the time required for onset of phlebitis following cannula insertion. This highlights the urgent need for further research not only on the incidence and predictors phlebitis but also on the timing of its onset, in order to inform evidence-based preventive strategies. Therefore, this study is significant in advancing clinical practice by determining the time to onset of phlebitis, estimating the incidence density rate of phlebitis, and identifying the predictors associated with time to onset of phlebitis among patients who were admitted with a peripheral intravenous cannula at DMCSH in 2025.

## Methods

### Study area and period

The study was conducted in the medical, surgical, and orthopedic wards of DMCSH. The hospital is located in Debre Markos, a metropolitan cite situated 295 km from Addis Ababa, the capital city of Ethiopia, and 265 km from Bahir Dar, the capital city of the Amhara regional state. DMCSH, is the only comprehensive specialized hospital in the area and provides healthcare services to a catchment area more than five million people. According to the hospital's annual report, the DMCSH recorded over 22,520 admissions in the last year, of which 6,976 patients were admitted to the three wards. Specifically, 472 patients were admitted to the medical ward, 584 to the surgical, and 96 to the orthopedic between January 1, 2024, and December 30, 2024. The data for this study was conducted from January 10, 2025, to April 12, 2025.

### Study design

This study utilized an institution-based prospective follow-up design.

### Population

**Source and study population.** The source population of this study consisted of all patients aged 15 years and above who were admitted to the medical, surgical, and orthopedic wards with a PIVC at DMCSH. Whereas, the study population included all patients aged 15 years and above who had a PIVC inserted during the study period in the same wards at DMCSH.

### Eligibility criteria

Patients aged 15 years and above who were admitted to the medical, surgical, and orthopedic wards and had a newly inserted PIVC during the study period at DMCSH were included in the study. However, patients with burns at PIVC insertion sites, those who had a PIVC inserted prior to the start of this study, and patients with altered consciousness were excluded from this study.

## Sample size and sampling technique

**Sample size determination.** The sample size for this study was calculated using Schoenfeld's for survival analysis [28]. A two-sided significance level (α) of 5% and a power of 80% were considered. The sample size was calculated after calculating the number of events as follow.

$$\text{Sample size (n)} = \frac{\text{Event}}{\text{probablity of event}}$$

$$\text{Event (E)} = \frac{(z\alpha/2 + Z\beta)^2}{pq(logHR)^2}$$

Event (E) $= \frac{4(z\alpha/2+Z\beta)^2}{(logHR)^2}$, under equal allocation of sample size in both groups (p=q)

Therefore, the overall sample size was calculated by the formula

$$\text{Event (E)} = \frac{4(1.96 + 0.84)^2}{(log0.36)^2} = \frac{31.36}{1.04} = 30$$

Therefore, the overall sample size was calculated by the formula

$$\text{Sample size (n)} = \frac{4(z\alpha/2 + Z\beta)^2}{(logHR)^2\Psi}$$

$$n = \frac{Event}{\Psi} = \frac{30}{\Psi}$$

n=$\frac{30}{0.09}$ =333. By adding a 10% non-response, the final maximum sample size was 367.

Where n is the sample size; Ψ = the overall probability of an event at the end of the study from previous similar studies (0.09), E = the total number of events, Zα/2 is a critical value which is 1.96; β is the power (80%), p is the proportion of population allocated for the first group, q proportion of population allocated for the second group and HR is the hazard ratio of each predictor were considered in the sample size calculation.

The sample size could also be determined using STATA version 17 with the command: power cox, hratio (….) failprob (….) wdprob (....). Using STATA, based on different predictors from previous studies; heparin [29], cannula dressing status [27], and administration of blood and drugs [27], were calculated as 372, 268 and 189, respectively. To ensure adequate power, the largest value, 372, was taken as final sample size.

## Sampling Techniques and Procedure

To select the study participants, the total admissions from the previous year over a two-month period in each ward were used as the sampling frame. Since the study was conducted in three wards, the sample size was proportional allocated to each ward based on number of patients admitted between January 10, 2024, to April 12, 2024. During this period, 472 patients were admitted to the medical ward, 584 to the surgical ward, and 96 to the orthopedic wards. Accordingly, proportional allocation yielded 152 participants for the medical, 188 for the surgical ward, and 32 for the orthopedic ward, giving a total sample size of 372.

A systematic random sampling method was utilized to select the study participants. The sampling interval(K) was determined by dividing the total admissions across the three wards during the reference period (N = 1152) by the required sample size(n = 372), yielding a value of approximately three. The first participant was selected by the lottery method, and subsequent participants were selected at every kth interval based on the order of their cannulation time.

## Study variables

**Dependent variable.** The dependent variable of this study was the time to phlebitis onset.

**Independent variables.** The independent variables were categorized into socio-demographic, maternal health, clinical, infusion-related and cannula-related factors. The socio-demographic predictors included age, sex, residence, occupation and level of education. Maternal health-related predictors consisted of contraceptive use and pregnancy status. Clinical predictors included body mass index, presence of chronic diseases, ambulation status, and admission diagnosis. Infusion-related predictors consisted of administration of fluids, drugs administration, blood transfusion, frequency of drug administration. Finally, cannula-related predictors included cannula size, cannula insertion site, and cannula dressing status.

## Operational definition

Phlebitis: phlebitis was assessed using internationally recognized Jackson's Visual Infusion Phlebitis (VIP) scoring system. A VIP score of zero indicates no phlebitis, a score of two marks the onset of phlebitis, a score of three indicates moderate phlebitis, a score of four represents advanced phlebitis, and a score of five reflects severe phlebitis or infection [27,30]. Therefore, in this study, phlebitis was considered as a VIP score of two or higher.

Event: The event was considered the development of phlebitis during the follow-up period.

Follow-up time: The follow-up time referred to the number of days from the insertion of PIVC until the occurrence of the event or censoring.

Censored: Patients were considered censored if they did not develop phlebitis during the study period due to reasons, such as removal of cannula before detection of phlebitis, lost follow-up, death, transfer, leaving against medical advice, or reaching the end of the follow-up period without phlebitis.

Cannula size: Cannula size was recorded according to the outer diameter of the needle, measured in French (Fr) units, with corresponding gauge and color codes: 16-gauge (gray, 5Fr), 18-gauge (green, 3.8Fr), 20-gauge (pink, 2.7Fr), 22-gauge (blue, 2.2Fr), 24 gauge (yellow, 1.7Fr) [16,31].

No ambulation: Patients were considered to have no ambulation if they were unable to cannot walk independently [32].

Appropriate cannula dressing status: Appropriate cannula dressing status was defined as a dressing that was clean, dry, and intact, with the cannula securely fixed and the insertion site covered over it [33].

Time to onset of phlebitis: The time to onset of phlebitis was defined as the duration from insertion of the cannula until phlebitis was detection.

Body mass index: Body mass index was calculated as weight in kilograms divided by height in meters squared and classified as underweight (<18.5), normal (18.5-23.99), overweight (24-27.99), obese (28 and above).

Admission diagnosis: The admission diagnosis was decided based on the primary clinical condition documented in the patient's medical record, which represented the main reason for hospitalization.

## Data collection tools

The data were collected using Kobo Toolbox. The data collection questionnaires were adapted from previously published literatures [7,12,16,22,27]. A pre-tested structured questionnaire, an observational checklist, and Jackson's VIP scoring system were used to collect the data. The questionnaire included socio-demographic factors, maternal factors,

clinical factors, cannula-related factors, infusion-related factors, and the outcome variable. The observation checklist also included the time of data collection in days.

Jackson's VIP scoring system was used to assessed the stage of phlebitis. A score of 0 indicated no sign of phlebitis, with the IV cannula site appearing health. A score of 1 (early sign) indicated either slight pain or redness near the IV site. A score of 2 (early stage) indicated the presence of two signs, namely pain, erythema, and swelling. A score of 3 (medium stage) indicated the presence of pain along the path of the cannula, erythema and induration. A score of 4 (advanced stage) indicated all of the following: pain along the path of the cannula, erythema, induration, and a palpable venous cord. A score of 5 (advanced stage/severe stage of thrombophlebitis) indicated the presence of pain along the path of the cannula, erythema, induration, a palpable venous cord, and pyrexia.

The data collection tools were initially prepared in English, translated into the local language (Amharic), and then back-translated into English to ensure consistency and accuracy.

## Data Quality Assurance

A pre-test was conducted on 5% of the sample size at DMCSH among patients with peripheral intravenous cannulas to assess the appropriateness of the data collection questionnaire and observational checklist. The pre-test revealed no need for modification regarding the clarity of the questions or the consistency of variable recording after reliability was conducted. To ensure accuracy and consistency, the questionnaire was translated from English into Amharic version and then back-translated into English which can further enhance the accuracy and consistency.

The principal investigator provided one day of training to data collectors and supervisors by focusing on the data collection tool and procedures, and even closely supervised and monitored the process to maintain data quality during the data collection period. The collected data were reviewed daily for completeness, and any challenges encountered during data collection were addressed accordingly. Finally, all data were checked for completeness and consistency during data management, storage, and analysis.

## Data collection procedure

After obtaining ethical clearance from Debre Markos University and permission from DMCSH, trained nurses were assigned to the study. Two BSc nurses were deployed to the emergency ward, and one BSc nurse was assigned to each the medical, surgical, and orthopedic wards. In addition, one MSc nurse was assigned as a supervisor to oversee the entire data collection process.

In the emergency ward the two data collectors registered the date and time of cannulation for patients who were later admitted to the inpatient wards and communicated this information to the respective ward data collectors. For patients admitted directly to inpatient wards without a peripheral intravenous cannula, the assigned nurse recorded the time of cannula insertion. Study participants were then selected using a systematic random sampling method, with every third eligible patient (k = 3) enrolled based on the admission.

Before data collection, patients were informed about the study's objectives, confidentiality, and their right to decline participation. Only those who provided consent were included. The selected patients were followed from the time of cannulation, and the cannulated vein was observed twice daily (morning and evening) by the assigned BSc nurses for signs of phlebitis, using Jackson's VIP scoring system. Each patient was followed for the duration of the first cannula dwelling time until they either developed phlebitis or were censored. Relevant medical history, clinically factors, and treatment-related information were extracted from patient charts and medication sheets, and nursing procedure records.

## Data management and analysis

Data were collected using Kobo Toolbox and exported to STATA version 17 for analysis. Prior to analysis, the dataset was checked for clarity, completeness, and consistency. Descriptive statistics, including frequencies, proportions, medians,

inter-quartile ranges, incidence, and incidence rates, were computed. The Kaplan-Meier method was used to estimate the cumulative probability of remaining phlebitis-free and to determine the median time to phlebitis onset. Differences in survival probabilities among categorical variables were assessed using the log-rank test. The association between independent predictors and the time to phlebitis onset was examined using the Cox proportional hazard regression model. Both bi-variable and multi-variable Cox regression analyses were performed.

The proportional hazards assumption was evaluated using Schoenfeld residuals test, interaction of covariates with time, and graphical methods. The goodness of fit for the model was evaluated through the Cox-Snell residual plot. The presence of multicollinearity among variables in the final fitted model was evaluated using the variance inflation factor. Predictor variables with a p-value <0.25 in the bi-variable analysis were included in the multi-variable model, and variables with a p-value <0.05 were considered independent predictors of phlebitis.

### Ethical consideration

Ethical clearance was obtained from Debre Markos University College of Medicine and Health Sciences Research Ethics and Approval Committee, with reference number RCSTTD/395/01/17, dated 01/11/2024. This study was conducted in accordance with the Declaration of Helsinki and relevant national guidelines for research involving human participants. Following ethical approval, additional written permission was obtained from DMCSH. Data collection was carried out in the medical, surgical, and orthopedic wards after all approvals were secured.

### Consent to participate

The objective and purpose of the study were clarified clearly to the patients, and verbal consent was obtained from each participant, with assent obtained from participants aged between 15–18 years. The Research Ethics and Approval Committee approved the use of verbal consent. Verbal consent was documented by the data collectors on the prepared checklist with his/her signature after the participant's agreement. Even, the consent process was witnessed by a ward nurse who are working in the hospital. The confidentiality of participants was secured by avoiding personal identifiers in the data collection tool. The participants were also informed that their involvement was voluntary and that they could withdraw at any time without affecting their care.

## Result

### Socio-demographic and reproductive health characteristics

A total of 372 patients participated in this study, yielding a 100% response rate. Among these, 204 (54.8%) were male. The median age of the participants was 35 years (inter-quartile range (IQR): 25–51 years), with ages ranging from 16 to 85 years. Two-thirds of the study participants (66.9%) were rural residents, and 215 (57.8%) were farmers. Among the 168 female participants, 11 (6.6%) were pregnant. Additionally, 45(26.8%) of the non-pregnant female participants reported using contraceptives, with 35 (20.8%) specifically using Depo-Provera (Table 1).

### Clinical, Infusion, cannula-related characteristics

Around 271 (72.9%) participants were admitted with a single diagnosis. Overall, 108(29.0%) participants had chronic diseases, of whom 38 (35.2%) had congestive heart failure. Regarding body mass index, 223(59.9%) of the participants had a normal body mass index, whereas 26 participants (7.0%) could not be categorized due to general body swelling, pregnancy, or disability.

Out of the total participants, 345 (92.7%) received intravenous drugs, and 192 participants (51.6%) received two or fewer injections per day. Whereas, 155 (41.7%) participants received intravenous fluids, and 37(10.0%) participants received blood transfusions. Cannula characteristics showed that 250 participants (67.2%) used an 18-gauge cannula,

**Table 1. Socio-demographic characteristics of patients admitted with intravenous cannula at DMCSH, 2025(N-372).**

| Variable | Categories | Total | |
|---|---|---|---|
| | | **Frequency** | **Percent** |
| Sex | Male | 204 | 54.84 |
| | Female | 168 | 45.16 |
| Age | 15-40 | 226 | 60.48 |
| | 41-60 | 85 | 22.85 |
| | >60 | 61 | 16.67 |
| Residence | Rural | 249 | 66.94 |
| | Urban | 123 | 32.16 |
| Occupational status | Farmer | 215 | 57.80 |
| | House wife | 73 | 11.83 |
| | Merchant | 44 | 10.75 |
| | Student | 40 | 19.62 |
| Educational status | Unable to read and write | 136 | 36.56 |
| | Informal school | 69 | 18.55 |
| | Elementary | 72 | 19.89 |
| | Secondary | 71 | 19.09 |
| | College and above | 22 | 24.81 |
| Pregnancy status | Yes | 11 | 6.55 |
| | No | 157 | 93.45 |
| Contraceptive use | Yes | 45 | 26.78 |
| | No | 123 | 73.21 |
| Type of contraceptive used | Depo-Provera | 35 | 20.83 |
| | Implanon | 10 | 5.95 |

and 212 (57.0%) had the cannula inserted in the forearm. Additionally, regarding cannula dressing status, most participants (313; 84.1%) had appropriate dressed cannula sites (**Table 2**).

### Incidence of Phlebitis

During the study period, 146 patients (39.3%; 95% CI: 34.25%–44.41%) developed phlebitis.

The overall incidence rate was 11 cases per 100 person-days of observation (95% CI: 9%–13%). Among those who develop phlebitis, majority had a VIP score of two Fig 1.

### Survival time of phlebitis onset

The total follow-up time for all participants was 1,3 person-days, with a median follow-up duration of four days (IQR: 3–4 days). The minimum and maximum follow-up time was one day and seven days respectively. The median time phlebitis onset was 5 days (IQR 4–6 days). Kaplan-Meier graph demonstrated that the probability of remaining phlebitis-free decreased as time progressed Fig 2.

### Survival difference between categories of predictors

Survival differences across categorical predictors were assessed using Kaplan-Meier curves and the log-rank test. Significant differences in time to phlebitis onset were observed for sex, blood transfusion, administration of vancomycin and KCl and cannula size.

**Table 2. Clinical, Infusion, cannula-related characteristics of patients admitted with intravenous cannula at DMCSH, 2025(N-372).**

| Variable | Categories | Total | |
|---|---|---|---|
| | | **FR** | **Percent** |
| Number of admission diagnoses | One | 271 | 72.85 |
| | >=two | 101 | 27.15 |
| Chronic diseases | Yes | 108 | 29.03 |
| | No | 264 | 70.97 |
| List of chronic diseases | Congested heart failure | 39 | 36.11 |
| | diabetes mellitus, | 15 | 13.89 |
| | Hypertension | 15 | 13.89 |
| | Asthma | 11 | 10.18 |
| | Retroviral infection | 10 | 9.26 |
| | Cancer | 7 | 6.46 |
| | Renal diseases | 7 | 6.46 |
| | Liver diseases | 6 | 5.55 |
| Body mass index | Not categorized | 26 | 6.99 |
| | Under weight | 83 | 22.31 |
| | Normal | 223 | 59.95 |
| | Over weight | 40 | 10.75 |
| Functional status | Ambulatory | 295 | 79.30 |
| | Not ambulatory | 77 | 20.70 |
| Fluid administration | Yes | 155 | 41.67 |
| | No | 217 | 58.33 |
| IV drug administration | Yes | 345 | 92.74 |
| | No | 27 | 7.26 |
| Number of injections per day | <=2 | 192 | 51.61 |
| | 3-4 | 64 | 17.20 |
| | >4 | 116 | 31.18 |
| Vancomycin | Yes | 53 | 14.25 |
| | No | 319 | 85.75 |
| Potassium chloride | Yes | 42 | 11.29 |
| | No | 330 | 88.71 |
| Anti-coagulant | Yes | 30 | 8.08 |
| | No | 342 | 91.94 |
| Blood transfusion | Yes | 37 | 9.95 |
| | No | 337 | 90.05 |
| Cannula size | 18 gauge | 250 | 67.21 |
| | 20 gauge | 56 | 15.05 |
| | 22 gauge | 66 | 17.74 |
| Cannula dressing status | Appropriate | 313 | 84.14 |
| | Not appropriate | 59 | 15.86 |
| Insertion site | Forearm | 212 | 56.99 |
| | Dorsum of hand | 77 | 20.70 |
| | Cubital fossa | 40 | 10.75 |
| | Palmar of the wrist | 43 | 11.56 |

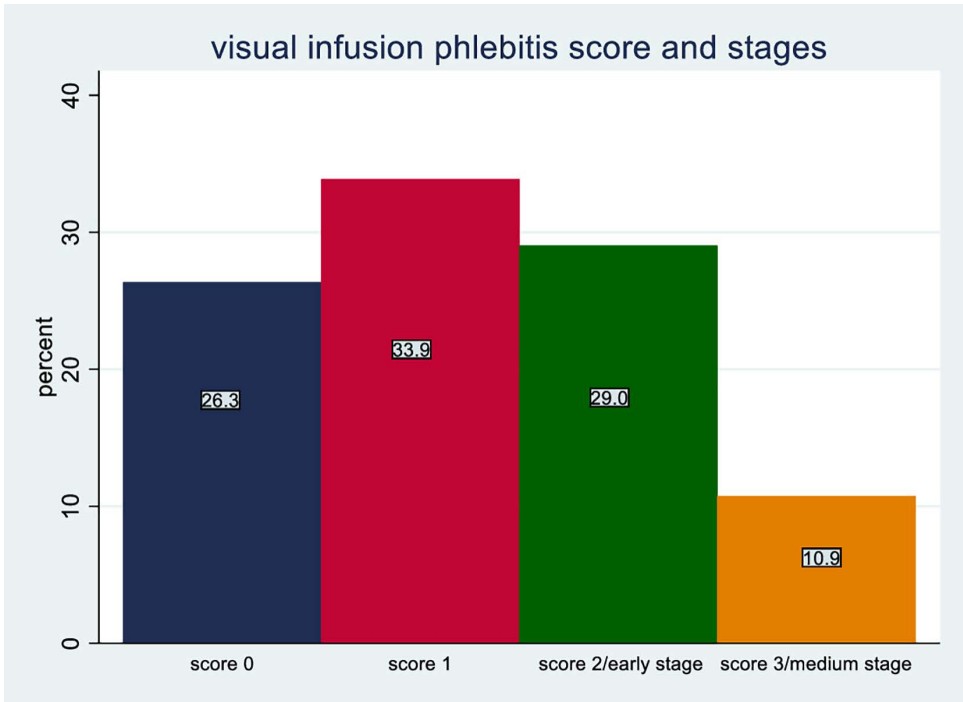

**Fig 1. Staging and scoring of Phlebitis among patients admitted with peripheral intravenous cannula at DMCSH, 2025.**

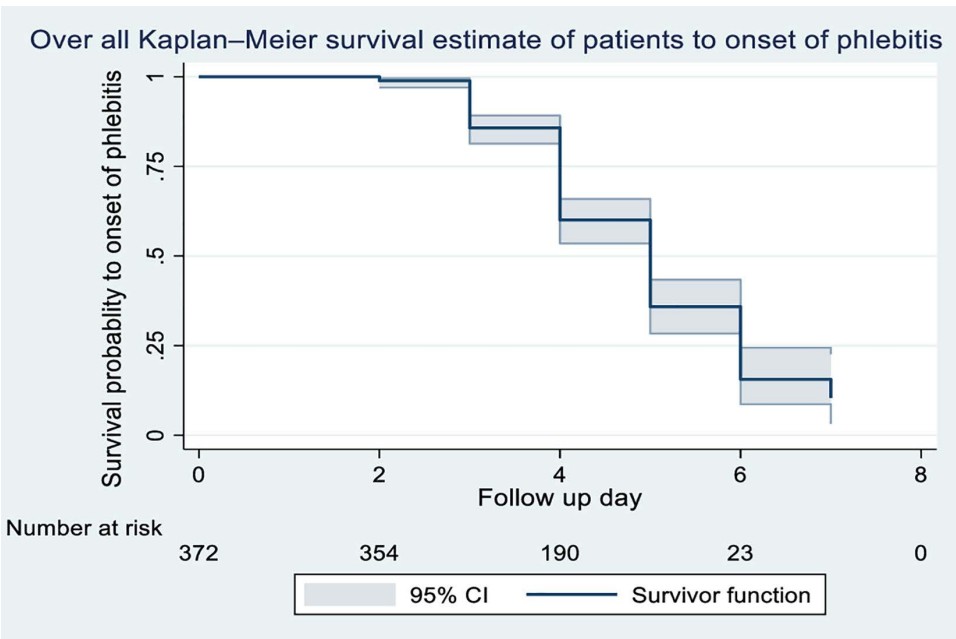

**Fig 2. Overall Kaplan-Meier survival probability of phlebitis onset among patients admitted with PIVC at DMCSH, 2025.**

According to the Kaplan-Meier survival curve and log-rank test, female patients had shorter median time to phlebitis onset (4 days) compared with males (5 days) (p=0.0013) Fig 3. The current study revealed that patients who received blood transfusions developed phlebitis (median: 3 days) compared with those with did not (median: 5 days) (p<0.001) Fig 4. Patients receiving vancomycin had a median time to phlebitis onset of 4 days versus 5 days for those not receiving vancomycin (p<0.001) Fig 5. Similarly, patients who received KCL had a median onset of 4 days compared with 5 days in those who did not (p<0.001) Fig 6. Cannulation with an 18G cannula was associated with a shorter median time to phlebitis onset (5 days) compared with other size (6 days) (p=0.0012) Fig 7.

## Proportional-hazard assumption test

The proportional-hazard assumption for variables included in the multi-variable Cox regression model was assessed using both statistical and graphical methods. The global Schoenfel residual test yielded p-value greater than 0.05 for all variables, with an overall global test value of 0.2674, indicating no violation of the proportional hazards assumption. Graphically, log-minus-log survival curves were plotted for categorical predictors, including sex, blood transfusion, and administration of vancomycin and KCL. The curves were approximately parallel over time, further supporting the fulfillment of the proportional assumption.

## Final model goodness of fit

The Cox-Snell residual plot showed that the cumulative hazard function closely followed the 45° reference line, indicating that the Cox proportional hazards model provided a good fit to the data Fig 8.

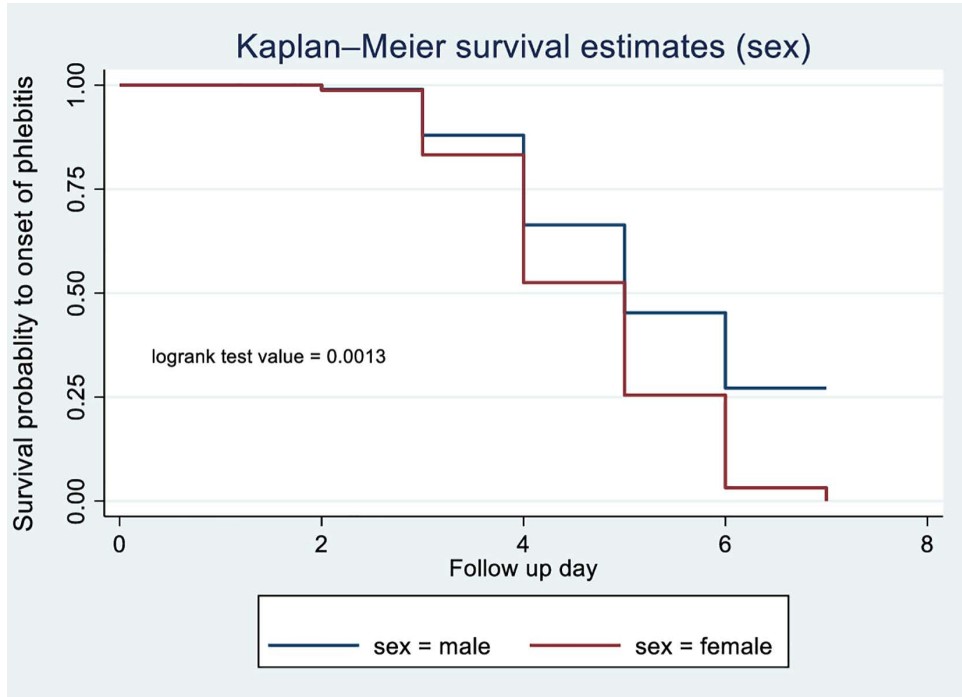

**Fig 3. Kaplan-Meier survival curve to compare survival difference of sex categories among patients admitted with PIVC at DMCSH, 2025.**

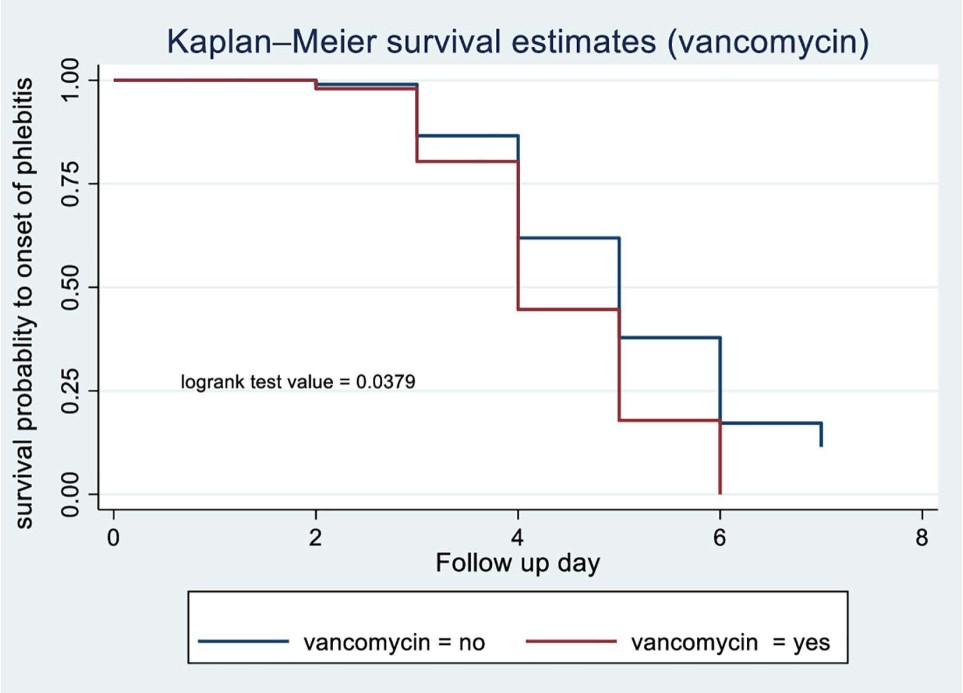

**Fig 4. Kaplan-Meier survival curve to compare survival difference of blood transfusion categories among patients admitted with PIVC at DMCSH, 2025.**

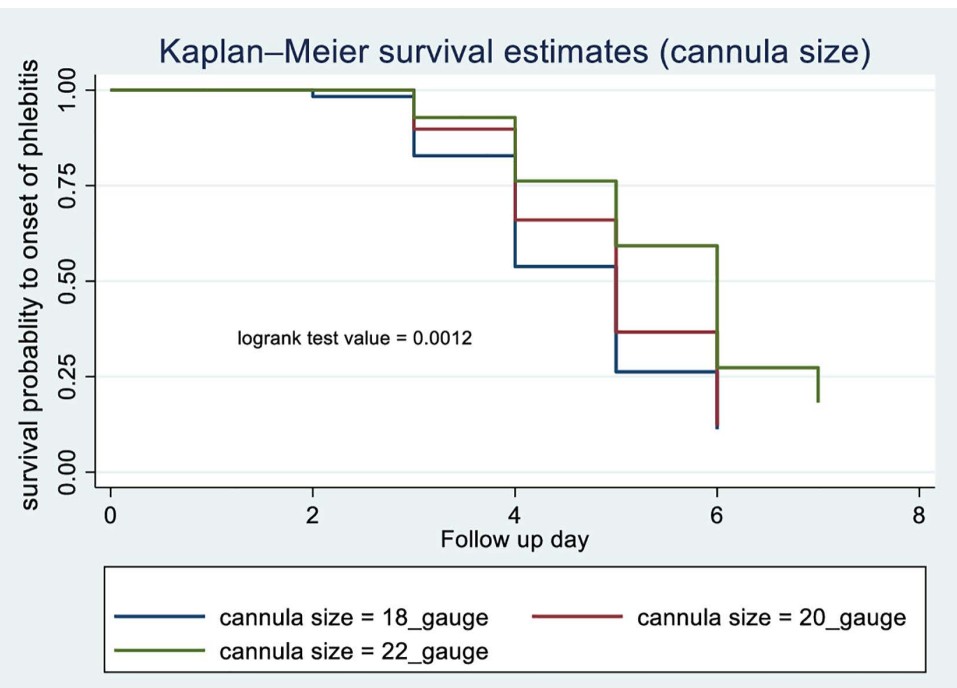

**Fig 5. Kaplan-Meier survival curve to compare survival difference of vancomycin administration categories among patients admitted with PIVC at DMCSH, 2025.**

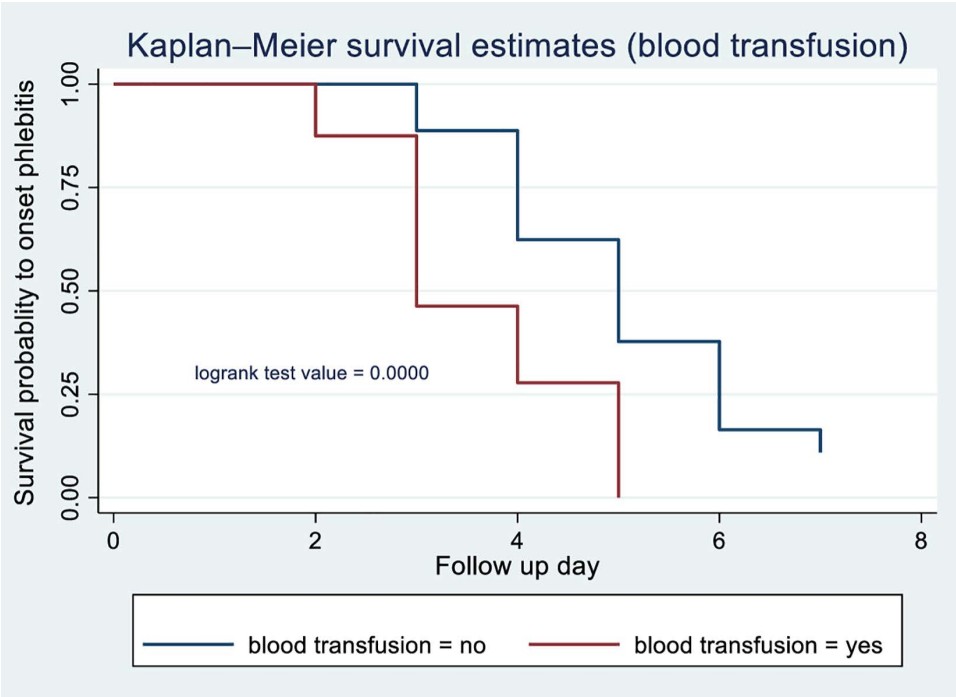

**Fig 6. Kaplan-Meier survival curve to compare survival difference of KCl administration categories among patients admitted with PIVC at DMCSH, 2025.**

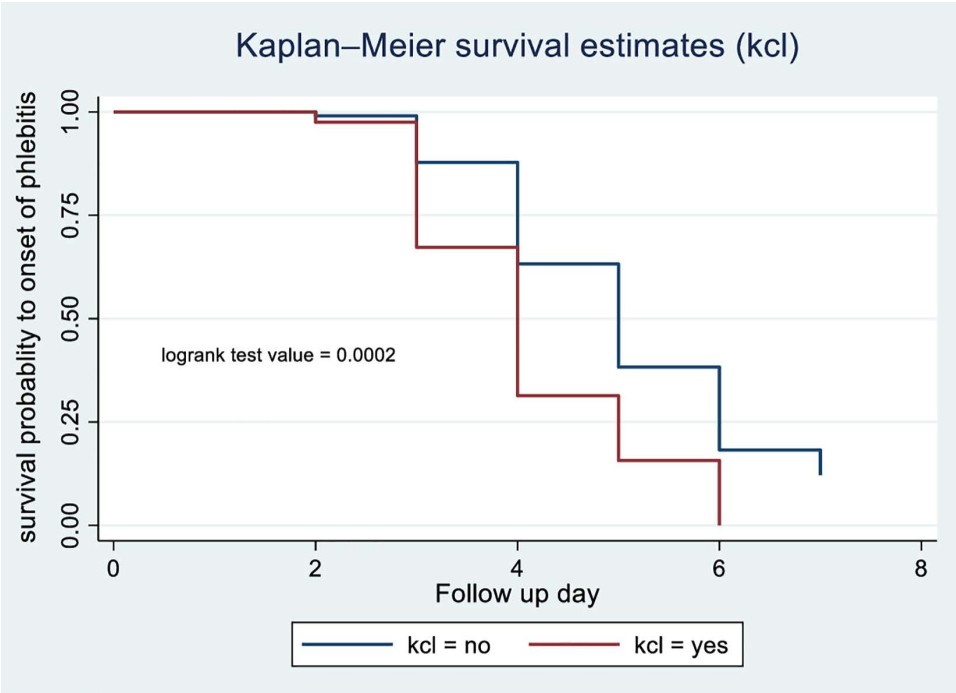

**Fig 7. Kaplan-Meier survival curve to compare survival difference of cannula size categories among patients admitted with PIVC at DMCSH, 2025.**

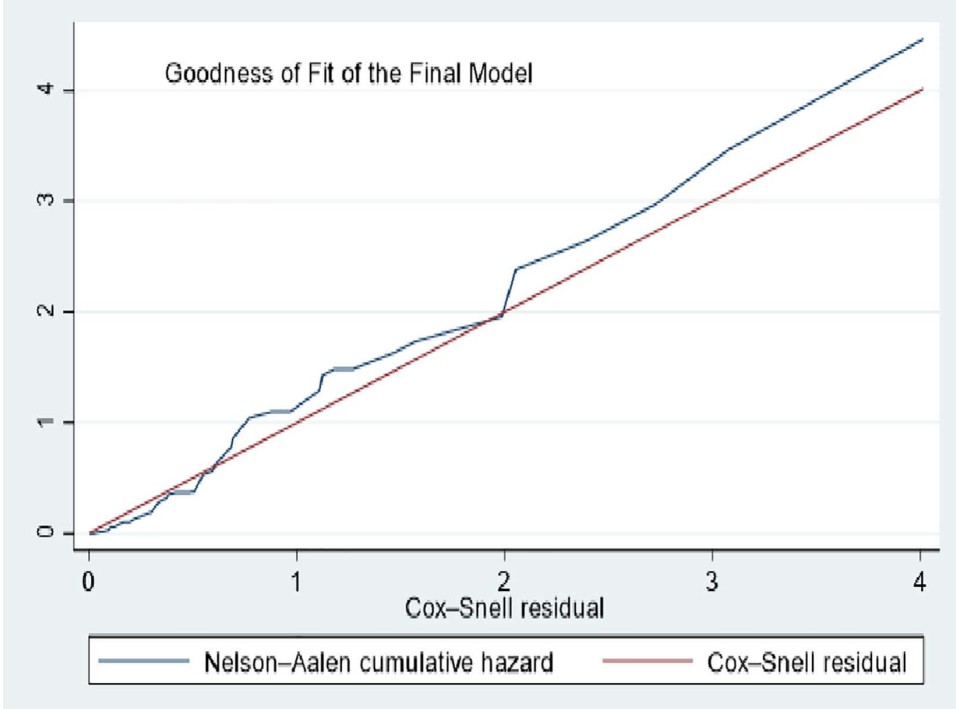

**Fig 8. Goodness-of-fit of the final Cox proportional hazards model using Cox-Snell residual among patients admitted with PIVC at DMCSH, 2025.**

## Predictors of phlebitis

Using bi-variable Cox regression analysis, twelve variables-including socio-demographic, clinical, fluid-related, and cannula-related predictors-were associated with phlebitis at a significance level of p < 0.25 and were included in the multi-variable regression model. In the final multi-variable Cox proportional hazards analysis, five predictors remained significantly associated with phlebitis (p < 0.05). These included female sex, blood transfusion, vancomycin administration, KCL administration, and cannula size.

The multi-variable Cox proportional regression analysis revealed that females had 58% higher hazard of developing phlebitis compared with male patients (AHR = 1.58; 95% CI: 1.11–2.25). Patients who received blood transfusions had a 2.11 times higher hazard of developing phlebitis than those who did not (AHR = 2.11; 95% CI: 1.14–3.91). Patients with an 18-gauge cannula had a 2.02 times higher hazard of phlebitis onset earlier comparted with those with a 22-gauge cannula (AHR = 2.02; 95% CI: 1.20–3.41). Patients who received vancomycin had a 1.93 times higher hazard of developing phlebitis compared with those did not (AHR = 1.93; 95% CI: 1.17–3.19). Lastly, those who received KCl had a 2.89 times higher hazard of developing phlebitis earlier than those who did not (AHR = 2.89; 95% CI: 1.73–4.83) (**Table 3**).

## Discussion

This study provides the first evidence on the time to onset of phlebitis and its predictors among admitted patients with a peripheral intravenous cannula at DMCSH. The findings showed admitted patients a cumulative incidence of 39.3% (95% CI 34.25% − 44.4%) and an incidence density rate of 11 cases per 100 persons-days of observation (95% CI 9% − 13%). These results are comparable to studies conducted in Bahir Dar, Ethiopia (cumulative incidence: 37.0%; incidence rate of

**Table 3. Bivariate and Multivariate Cox regression analysis for the predictors of phlebitis among admitted patients with PIVC in DMCSH, 2025(N-372).**

| List of variables | category | Phlebitis | | CHR(95% CI) | AHR(95% CI) | p-value |
|---|---|---|---|---|---|---|
| | | Event | Censored | | | |
| Sex | Male | 66 | 138 | 1 | 1 | |
| | Female | 80 | 88 | 1.57(1.13,2.17) | 1.58(1.11,2.25) | **0.010** |
| Age | >60 | 25 | 36 | 1 | 1 | |
| | 15-40 | 31 | 54 | 1.53(.98, 2.40) | 1.44(0.91,2.45) | 0.109 |
| | 41-60 | 90 | 136 | 1.21(.70, 2.05) | 1.11(0.64,1.92) | 0.701 |
| Chronic diseases | No | 93 | 171 | 1 | 1 | |
| | Yes | 53 | 55 | 1.24(.88, 1.73) | 1.31(0.90,1.91) | 0.159 |
| Fluid administration | No | 93 | 124 | 1 | 1 | |
| | Yes | 53 | 102 | 0.80(.57, 1.13) | 0.76(0.52,1.10) | 0.146 |
| Drug administration | No | 10 | 33 | 1 | 1 | |
| | Yes | 136 | 193 | 1.6(.84, 3.03) | 1.21(0.63,2.35) | 0.566 |
| Blood transfusion | No | 131 | 204 | 1 | 1 | |
| | Yes | 15 | 22 | 3.7(2.15, 6.43) | 3.40(1.85,6.29) | **0.000** |
| Anticoagulant administration | No | 131 | 211 | 1 | 1 | |
| | Yes | 15 | 15 | 0.69(.40, 1.18) | 0.69(0.38,1.23) | 0.207 |
| Cannula size | 18 gauge | 101 | 149 | 1.98(1.25,3.13) | 1.91(1.13,3.21) | **0.015** |
| | 20 gauge | 20 | 36 | 1.45(.79, 2.65) | 1.28(0.68,2.39) | 0.440 |
| | 22 gauge | 25 | 41 | 1 | 1 | |
| Insertion site | Forearm | 81 | 131 | 1 | 1 | |
| | Dorsum of wrist | 25 | 52 | 0.97(.62, 1.53) | 1.03(0.62,1.70) | 0.920 |
| | Cubital fossa | 21 | 19 | 1.26(.78, 2.04) | 1.44(0.87,2.40) | 0.159 |
| | Palmar of wrist | 19 | 24 | 1.35(.82, 2.24) | 1.45 (.86, 2.46) | 0.165 |
| Cannula dressing status | Appropriate | 122 | 191 | 1 | 1 | |
| | Not appropriate | 24 | 35 | 1.4(.90,2.16) | 1.13(.76, 1.98) | 0.400 |
| KCL | No | 123 | 207 | 1 | 1 | |
| | Yes | 23 | 19 | 2.03(1.29,3.17) | 2.28(1.39,3.74) | **0.001** |
| Vancomycin | No | 126 | 193 | 1 | 1 | |
| | Yes | 20 | 33 | 1.54(.96, 2.48) | 1.75(1.06,2.88) | **0.029** |

Foot note: The bold P values are statistically significant predictors with p value less than 0.05, CHR; crude hazard ratio and AHR; Adjusted hazard ratio

8 per 100 person-day) [27], Tunisia (cumulative incidence: 33.3%; incidence rate: 8.8 per 100 cannula days) [34], and in Pakistan (cumulative incidence: 39.1%) [22].

However, the incidence observed in this study was higher than reported in Japan, where a multicenter ICU-based cohort study documented 9.1% incidence and 3.5 cases per 100 person-days [19]. lower figures were also reported in Spain (incidence: 10.68%; incidence density rate: 1.8 cases per 100 person-days) [35] and Brazil (incidence: 7.2%: incidence density rate: 6.1% cases per 100 person-days) [20]. Similarly, studies in India(31.4%) [9] and Turkey (29.2% and 31%) [3,21] reported lower proportions. These discrepancies could be attributed differences in study stetting, sample sizes, and quality of care. For instance, the Japanese and Turkish studies were conducted in ICUs, where patients receive more intensive monitoring and holistic care, potentially lowering phlebitis risk. In contrast, the Brazilian study involved a small sample size (165 participants), that may have underestimate the true burden of phlebitis. Even it may relate with high nurse-to-patient ratios, lack of standardized IV protocols, and socio-economic difference.

Conversely, the incidence in this study was lower compared to that of Ghana (52.2%) [36] and the university of Gonder hospital (70%) [16]. The Ghanaian study included younger participants (10–15 years), who may mount stronger inflammatory responses, increasing the risk of phlebitis. Meanwhile, the Gondar study used a non-probability sampling technique, which may have overestimated the true figure.

The median time phlebitis onset in this study was five days, (IQR: 4–6 days), while the median cannula dwelling time was four days (IQR: 3–4 days). These findings consistent those from Bahir Dar (median dwell time: 4 days; median onset 6 days) [27], and Ghana (mean dwell time: 3.94 days) [36]. However, the mean follow-up time in this study was shorter than in Tunisia, where it was five days [34]. The possible discrepancy may be due to socio-economic difference and study conducted wards difference.

In this study, female patients had a higher hazard of developing phlebitis earlier compared to male. This finding is consistent with studies conducted in Ethiopia, India, other countries [2,9,16,37]. The likely explanation is the influence of estrogen on coagulation pathways, which elevates prothrombotic factors and predisposes females to phlebitis [38].

Blood transfusion was also identified as significant predictor of phlebitis, consistent with findings Ethiopia [27], India [9] and Egypt [39]. The mechanism behind this association may be attributed to several factors. Blood products have higher viscosity and variable pH compared to IV fluids, potentially causing endothelial irritation. Moreover, prolonged transfusion times and transfusion-related reactions may contribute to vascular irritation and inflammation [40]. Similarly, KCl administration was strongly associated with phlebitis, echoing findings from India [9], China [41], and Thailand [42]. This is more likely due to KCl's high osmolarity, which causes osmotic fluid shifts and endothelial cell dehydration, resulting to vascular irritation [43]. Vancomycin administration was another significant risk factor, as supported by studies in India, [9] and Italy [44]. It is more probably due to a low Ph, high osmolarity, and prolonged infusion times can predispose peripheral veins to irritation [43,45].

The size of cannula also played a crucial role in the onset of phlebitis. Patients with 18-gauge cannulas developed phlebitis than those with 22-gauge cannulas, consistent with evidence from Ethiopia [27] and Spain [35] Larger cannula diameters increase mechanical irritation and direct endothelial injury, increasing the likelihood of phlebitis [46]. Even, it strongly recommends to adhere to evidence-based guidelines such as the infusion therapy standards of practice (9th edition). Preventive strategies should include regular site assessment, timely cannula replacement, and ongoing staff training. Cannula should be replaced based on clinical indication rather than routine time intervals and removed immediately at the first signs of phlebitis. Furthermore, the smallest possible gauge should be used that accommodates prescribed therapy, and cannulation of flexion sites should be avoided unless clinically justified [2,46,47].

### Limitations of the study

This study was conducted in a single-center setting and focused only in three words, which may limit the generalizability of the findings. Observational design may slightly restrict the ability to precisely identify the main outcome. The nature of observational design may restrict the ability to establish causal relationships between the identified factors and the onset of phlebitis. There is a possibility of residual confounding that cannot be fully excluded. Although pregnancy status and contraceptive use were initially considered, analysis could not be conducted due to incomplete data. These limitations restrict the ability to draw conclusions on maternal factors may influence the risk of phlebitis. Lastly, there may be a potential of risk of hidden periodicity bias when systematic sampling applied.

### Conclusion and recommendation

#### Conclusion

This study showed a relatively high cumulative incidence and incidence density of PIVC, with a notably shorter median time to phlebitis onset among admitted patients. Significant predictors of phlebitis onset included female sex, use of an 18-gauge cannula, receipt of blood transfusion, and administration of KCL and vancomycin.

## Recommendation

To handle the risk of phlebitis, healthcare providers should implement targeted prevention strategies, particularly for high-risk groups such as female patients and those receiving blood transfusions, KCL and vancomycin. Whenever clinically feasible, the use of smaller-gauge cannulas is recommended to minimize direct endothelial damage of venous walls. Also, irritant medications such as vancomycin and KCl should be diluted and administered slowly with frequent monitoring of the IV site to minimize chemical irritation of venous walls.

Whereas, at the institutional level such as healthcare organizations should establish and enforce standardized protocols that incorporate the identified risk factors in the above. These should also include ensuring the availability of smaller gauge cannulas and providing training particularly focusing on high-risk drugs.

Future studies are warranted to include the limitations of this study. Specifically, multiple center studies are needed to enhance the generalizability of the findings and to explore potential maternal related factors contributing to the higher risk observed among female patients. Such investigations will provide more comprehensive evidence to policy in reducing the burden of phlebitis.

## Author contributions

**Conceptualization:** Zinaw Beyene, Haymanot Zeleke Mitiku, Hailie Amha.

**Data curation:** Zinaw Beyene, Haymanot Zeleke Mitiku, Tesfahun Ayenew, Mengistu Abebe, Mihret Kefie, Afework Edmealem, Hailie Amha.

**Formal analysis:** Zinaw Beyene, Haymanot Zeleke Mitiku, Tesfahun Ayenew, Mengistu Abebe, Mihret Kefie, Afework Edmealem, Hailie Amha.

**Funding acquisition:** Zinaw Beyene, Hailie Amha.

**Investigation:** Zinaw Beyene, Haymanot Zeleke Mitiku, Mengistu Abebe.

**Methodology:** Zinaw Beyene, Haymanot Zeleke Mitiku, Tesfahun Ayenew, Mengistu Abebe, Mihret Kefie, Afework Edmealem, Hailie Amha.

**Resources:** Zinaw Beyene.

**Software:** Tesfahun Ayenew.

**Supervision:** Haymanot Zeleke Mitiku, Afework Edmealem, Hailie Amha.

**Validation:** Haymanot Zeleke Mitiku, Tesfahun Ayenew, Mengistu Abebe, Afework Edmealem, Hailie Amha.

**Visualization:** Haymanot Zeleke Mitiku, Mengistu Abebe, Afework Edmealem, Hailie Amha.

**Writing – original draft:** Zinaw Beyene, Tesfahun Ayenew.

**Writing – review & editing:** Haymanot Zeleke Mitiku, Tesfahun Ayenew, Mengistu Abebe, Mihret Kefie, Afework Edmealem, Hailie Amha.

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
