## [Decision Letter · Decision Letter 0]

2 Sep 2025

Dear Dr. Mitiku,

Thank you for submitting your manuscript to PLOS ONE. After careful consideration, we feel that it has merit but does not fully meet PLOS ONE’s publication criteria as it currently stands. Therefore, we invite you to submit a revised version of the manuscript that addresses the points raised during the review process.

We look forward to receiving your revised manuscript.

Kind regards,

Federica Canzan

Academic Editor

PLOS ONE

Journal Requirements:

2. In the ethics statement in the Methods, you have specified that verbal consent was obtained. Please provide additional details regarding how this consent was documented and witnessed, and state whether this was approved by the IRB

3. Please include captions for your Supporting Information files at the end of your manuscript, and update any in-text citations to match accordingly. Please see our Supporting Information guidelines for more information: http://journals.plos.org/plosone/s/supporting-information .

Reviewers' comments:

Reviewer's Responses to Questions

**Comments to the Author**

1. Is the manuscript technically sound, and do the data support the conclusions?

Reviewer #1: Yes

Reviewer #2: Yes

2. Has the statistical analysis been performed appropriately and rigorously?

Reviewer #1: Yes

Reviewer #2: Yes

3. Have the authors made all data underlying the findings in their manuscript fully available?

Reviewer #1: Yes

Reviewer #2: Yes

4. Is the manuscript presented in an intelligible fashion and written in standard English?

Reviewer #1: No

Reviewer #2: Yes

Reviewer #1: The manuscript titled “Time to develop phlebitis and its predictors among admitted patients with peripheral intravenous cannulas at Debre Markos Comprehensive Specialized Hospital, Northwest Ethiopia, 2024/2025: a prospective follow-up study” addresses an important topic in patient safety and intravenous therapy management. The study provides valuable data on the incidence and predictors of phlebitis in an Ethiopian hospital setting, which is a relatively under-researched area. Overall, the manuscript is informative and well-structured. However, there are several points regarding clarity, methodology, data presentation, discussion, and ethical considerations that should be addressed to strengthen the manuscript.

Abstract

Background: Some sentences in the background (line 33-37) repeat ideas about phlebitis being a clinical problem without focusing directly on the research gap. So please write the background to highlight the knowledge gap in Ethiopia more clearly.

Methods: It is good that statistical methods (Kaplan–Meier, Cox regression) are mentioned, but they could briefly state why survival analysis was chosen (time-to-event nature).

Result: the results are well presented with cumulative incidence and median time to event. It may help readers if confidence intervals are also provided for incidence and survival estimates, not just hazard ratios.

Conclusion: The conclusion is consistent with the results. However, the authors might phrase recommendations for clinical practice or implications more directly (e.g., “Clinicians should monitor patients with 18-gauge cannulas or receiving vancomycin more closely…”).

Please consider adding a brief statement on the major limitation of the study in the abstract (e.g., sample size, study design), as this is important for balance and clarity.

Introduction

The manuscript currently separates Background and Objectives into two distinct sections. In journal article format, it is preferable to combine these into a single Introduction section. The introduction should begin with a concise overview of the clinical relevance of peripheral intravenous cannula (PIVC) and phlebitis. Summarize global and local incidence rates, highlighting the gap in Ethiopian data. End with a clear statement of the study objectives.

The background is informative but overly detailed in places, which may obscure the main research gap. For instance, lines 62–64 include a long sentence with multiple procedures listed; this could be simplified for readability.

Consider tightening the section by focusing less on general definitions and more on why phlebitis is a pressing issue in Ethiopia specifically.

Line 61: “Each year, over one billion intravenous cannulas are placed for patients in hospitals patients globally” → appears redundant (“patients” twice).

Line 76: “It may be sue to fail to adhere” → should be “due to failure to adhere.”

Ensure terminology is consistent (e.g., “predictors” vs. “predicators” in line 106).

The background cites global incidence (3–70%), with examples from multiple countries, which is strong. However, it may help to briefly note why Ethiopia shows such a high rate (70% in Gondar hospital). Is it due to methodological differences, healthcare infrastructure, or clinical practices?

Line 70: Suggest rephrasing “Among mechanical factors, the size of the cannula and the site of insertion play a significant role” → could be “Mechanical factors such as cannula size and site of insertion significantly influence risk.”

Formatting: Citations are in parentheses but inconsistent spacing (e.g., “clinical setting(1)” → should be “clinical setting (1)”).

Methods

The methodology is described in detail, which is commendable. However, the section is very long and could benefit from more concise phrasing. Some repetition (e.g., in sample size determination and cannula definitions) could be reduced for clarity.

Study area and period: The study area is limited to medical, surgical, and orthopedic wards. However, important clinical areas such as maternal, or ICU wards were not included. This restriction may affect the generalizability of the findings, as the risk of phlebitis and related predictors could differ across different patient populations. I recommend that the authors acknowledge this limitation, particularly in the abstract and discussion sections.

Study design and population: The design (prospective follow-up) is appropriate for the research objective.

Inclusion and exclusion criteria are well stated. However, justification for excluding patients with altered consciousness could be clarified (e.g., inability to provide consent, difficulty in observing outcomes).

Sample size and sampling technique: The proportional allocation and systematic sampling are appropriate, though the potential limitation of systematic sampling (risk of hidden periodicity bias) should be acknowledged.

Data collection: It would help to specify whether inter-rater reliability was checked among data collectors when assessing phlebitis scores, to strengthen methodological rigor.

Data quality assurance: Translation and back-translation are appropriate, but a note on whether content validity or reliability testing of the questionnaire was performed would strengthen this section.

Data management and analysis: Testing of proportional hazards assumptions and checking for multicollinearity are strengths. However, the authors should mention how missing data (if any) were handled.

Ethical consideration: The manuscript mentions that ethical approval was obtained, but the reference number or formal approval letter is not provided. For transparency and compliance with journal standards, it is recommended to include the ethics committee approval reference number and, if available, the date of approval. This ensures readers can verify that the study was conducted in accordance with ethical guidelines.

Result

Socio-demographic characteristics: The reporting of contraceptive use could be clarified further; Is the denominator all female participants (168), or only non-pregnant ones?

Clinical, Infusion, cannula-related characteristics: Nearly three-fourths of the study participants were admitted with a single diagnosis” – It would be clearer to provide the exact number and percentage instead of a vague phrase (“nearly three-fourths”).

The phrase “three-fourths of the participants had a normal BMI” is too approximate. Giving exact numbers and percentages would be more precise.

Treatment Data (IV fluids, drugs, injections, blood transfusion): The flow is a bit disorganized. It mixes different interventions without grouping them logically. Consider restructuring: first fluids/drugs, then injection frequency, then transfusion.

Almost half patients (51.61%) received two or fewer injections per daily” → needs correction for grammar: “…per day.

Again, percentages like 29.03%, 35.19%, 41.67%, etc. are overly precise. Rounding to one decimal place (e.g., 29.0%, 35.2%, 41.7%) is more standard in reporting.

Predictors of phlebitis: Use either “Cox proportional hazards regression” or “Cox regression” consistently. Mixing terms (“Cox proportional hazards model”, “Cox proportional regression”) may confuse readers.

Discussion

This is a strong discussion with thorough literature engagement and plausible explanations, but it is too wordy, repetitive, and detail-heavy. It would benefit from reorganization, clearer writing, and more emphasis on implications.

The discussion provides a strong comparison with previous studies; however, it would benefit from additional justification for the high incidence of phlebitis in your setting. Beyond methodological and demographic explanations, systemic factors in Ethiopia (such as limited resources, high nurse-to-patient ratios, lack of standardized IV protocols, and socio-economic challenges) may have contributed.

Recommendations

For health organizations: Consider rephrasing “ensure availability of alternative medications to Vancomycin and KCl” alternatives may not always exist; instead, focus on safe administration practices, staff training, and supply of appropriate equipment.

Limitation of the study

Expand slightly to include other study limitations, e.g., single-center design, observational nature, and potential for residual confounding.

Reviewer #2: Summary of Reviewer Comments

Abstract Section

Define outcome variables like incidence of phlebitis and time to onset for clarity.

Methods

Clarify the choice of median over mean.

Results Section

Specify if pregnancy status and contraceptive use are sociodemographic or health-related variables for better categorization.

Main Body

VIP Scoring System: The lack of a score of one may limit early detection of phlebitis. Consider adding this score to improve sensitivity and clinical assessments.

Discussion Section

Emphasize the study's implications for patients, healthcare professionals, and policymakers rather than just comparing findings.

**Do you want your identity to be public for this peer review?** For information about this choice, including consent withdrawal, please see our Privacy Policy

Reviewer #1: No

Reviewer #2: No

---

## [Author Response · Author response to Decision Letter 1]

1 Oct 2025

We would like to say thank you (Federica Canzan- academic editor) for fast response for our submission.

Also, we would like to thank the reviewers for sacrificing their valuable time and providing constructive comments on this study. We have made all corrections in the main document, highlighted in yellow for reviewer one, purple color for reviewer two and green for editor based on their feedback in the main document which is named revised manuscript with change.

Editor comments

Response

We used PLOS ONE's style requirements, including those for file naming.

2. In the ethics statement in the Methods, you have specified that verbal consent was obtained. Please provide additional details regarding how this consent was documented and witnessed, and state whether this was approved by the IRB

Response

Thank you, we modified this in the revised manuscript with retract changed that highlighted by green color.

3. Please include captions for your Supporting Information files at the end of your manuscript, and update any in-text citations to match accordingly.

Response

We changed the supporting information by narration, which was highlighted in green color in the revised manuscript with retract changed.

Response

We carefully examined the paper suggested by the reviewers. After evaluating their relevance to our study, we incorporated the works that were directly pertinent and strengthened our paper. For those that were less important to our paper, we did not add them.

Response

Thank you for your valuable comments. We carefully reviewed our reference list to ensure accuracy and completeness. During this process, we identified and replaced some older references with more recent and relevant papers. None of the cited articles are retracted; therefore, no retraction notices were required. All changes to the reference list have been reflected in the revised manuscript.

Reviewer's One Questions and Comments:

We would like to thank the reviewer for sacrificing your valuable time and providing constructive comments on this study. We have made all corrections in the main document, highlighted in yellow, based on your feedback. However, corrections highlighted in pink were made for Reviewer Two. Thank you both

Reviewer’s Q:

The manuscript titled “Time to develop phlebitis and its predictors among admitted patients with peripheral intravenous cannulas at Debre Markos Comprehensive Specialized Hospital, Northwest Ethiopia, 2024/2025: a prospective follow-up study” addresses an important topic in patient safety and intravenous therapy management. The study provides valuable data on the incidence and predictors of phlebitis in an Ethiopian hospital setting, which is a relatively under-researched area. Overall, the manuscript is informative and well-structured. However, there are several points regarding clarity, methodology, data presentation, discussion, and ethical considerations that should be addressed to strengthen the manuscript.

Author's Response:

Thank you very much for your constructive and insightful feedback on this paper. We greatly appreciate your recognition to the importance of our research work. We carefully reviewed all the points you mentioned regarding to the clarity, methodology, data presentation, discussion, and

We carefully reviewed all the points you raised regarding clarity, methodology, data presentation, discussion, and ethical considerations, and we have made the necessary corrections and improvements throughout the manuscript.

Reviewer’s Q:

Abstract

Background: Some sentences in the background (line 33-37) repeat ideas about phlebitis being a clinical problem without focusing directly on the research gap. So please write the background to highlight the knowledge gap in Ethiopia more clearly.

Methods: It is good that statistical methods (Kaplan–Meier, Cox regression) are mentioned, but they could briefly state why survival analysis was chosen (time-to-event nature).

Result: the results are well presented with cumulative incidence and median time to event. It may help readers if confidence intervals are also provided for incidence and survival estimates, not just hazard ratios.

Conclusion: The conclusion is consistent with the results. However, the authors might phrase recommendations for clinical practice or implications more directly (e.g., “Clinicians should monitor patients with 18-gauge cannulas or receiving vancomycin more closely…”).

Please consider adding a brief statement on the major limitation of the study in the abstract (e.g., sample size, study design), as this is important for balance and clarity.

Author's Response:

Thank you for your valuable comment. We did an update on the abstract thanks highlighted in yellow color

Reviewer’s Q:

Introduction

The manuscript currently separates Background and Objectives into two distinct sections. In journal article format, it is preferable to combine these into a single Introduction section. The introduction should begin with a concise overview of the clinical relevance of peripheral intravenous cannula (PIVC) and phlebitis. Summarize global and local incidence rates, highlighting the gap in Ethiopian data. End with a clear statement of the study objectives.

Author's Response:

Thanks, we corrected it.

Reviewer’s Q:

Introduction

The background is informative but overly detailed in places, which may obscure the main research gap. For instance, lines 62–64 include a long sentence with multiple procedures listed; this could be simplified for readability.

Author's Response:

lines 62–64:- While it aids in medication administration, fluid infusion, blood transfusion, and supports numerous therapeutic and diagnostic procedures like hemodialysis, parenteral nutrition, and contrast agent delivery, patient may experience complications like phlebitis

changed by

While this procedure is essential for giving medications, fluid, blood transfusions, and supporting treatments and diagnostic procedures like hemodialysis, parenteral nutrition, and contrast studies, it can also cause complications, the most frequent of which is phlebitis.

This keeps the contrast clear and makes it flow as a single complete sentence. We would like to appreciate the concern.

Reviewer’s Q:

Line 61: “Each year, over one billion intravenous cannulas are placed for patients in hospitals patients globally” → appears redundant (“patients” twice).

Author's Response: Thanks, corrected.

Reviewer’s Q:

Line 76: “It may be sue to fail to adhere” → should be “due to failure to adhere.”

Author's Response:

Thanks, corrected.

Reviewer’s Q:

Ensure terminology is consistent (e.g., “predictors” vs. “predicators” in line 106).

Author's Response:

Thanks, corrected.

Reviewer’s Q:

The background cites global incidence (3–70%), with examples from multiple countries, which is strong. However, it may help to briefly note why Ethiopia shows such a high rate (70% in Gondar hospital). Is it due to methodological differences, healthcare infrastructure, or clinical practices?

Author's Response:

Thank you for the insightful comment. The high incidence reported in Gondar Hospital (70%) may be attributed to factors such as differences in study methodology, variations in healthcare infrastructure, and specific clinical practices, including frequent use of peripheral intravenous cannulae and differences in infection prevention measures. To maintain clarity and conciseness in the background, we focused on reporting the incidence, while these potential contributing factors are discussed in detail in the discussion section.

Reviewer’s Q:

Line 70: Suggest rephrasing “Among mechanical factors, the size of the cannula and the site of insertion play a significant role” → could be “Mechanical factors such as cannula size and site of insertion significantly influence risk.”

Author's Response:

Several factors can contribute for occurrence of phlebitis. These are mechanical, chemical, and bacterial causes. Mechanical factors, such as cannula size and insertion sit, can irritate the vein wall and trigger inflammation. Chemical factors also contribute, particularly the type of solution and the infusion speed, with the faster rates increasing the likelihood of vein irritation. In addition, bacterial factors play a critical role, as improper sterile technique during cannula insertion can enter bacteria into the bloodstream, resulting to infection.

Reviewer’s Q:

Formatting: Citations are in parentheses but inconsistent spacing (e.g., “clinical setting(1)” → should be “clinical setting (1)”).

Author's Response:

• Thank you.

Reviewer’s Q:

Methods

The methodology is described in detail, which is commendable. However, the section is very long and could benefit from more concise phrasing. Some repetition (e.g., in sample size determination and cannula definitions) could be reduced for clarity.

Author's Response:

We used more concise phrase in the main document particularly in method section. Fore example, cannula definition was done as follow

Cannula size was recorded according to the outer diameter of the needle, measured in French (Fr) units, with corresponding gauge and color codes: 16-gauge (gray, 5Fr), 18-gauge (green, 3.8Fr), 20-gauge (pink, 2.7Fr), 22-gauge (blue, 2.2Fr),24 gauge (yellow, 1.7Fr).

Reviewer’s Q:

Study area and period: The study area is limited to medical, surgical, and orthopedic wards. However, important clinical areas such as maternal, or ICU wards were not included. This restriction may affect the generalizability of the findings, as the risk of phlebitis and related predictors could differ across different patient populations. I recommend that the authors acknowledge this limitation, particularly in the abstract and discussion sections.

Author's Response:

• Thank you for the constructive feedback. We acknowledge it in the limitation section

Reviewer’s Q:

Study design and population: The design (prospective follow-up) is appropriate for the research objective.

Author's Response:

• Okay, thank you, we wrote based on this.

Reviewer’s Q:

Inclusion and exclusion criteria are well stated. However, justification for excluding patients with altered consciousness could be clarified (e.g., inability to provide consent, difficulty in observing outcomes).

Author's Response:

• Thank you for the constructive feedback. Patients with altered consciousness were excluded primarily because they were unable to provide informed consent and their condition made it difficult to reliably assess and observe the study outcomes. This exclusion was necessary to ensure both ethical considerations and the accuracy of data collection

Reviewer’s Q:

Sample size and sampling technique: The proportional allocation and systematic sampling are appropriate, though the potential limitation of systematic sampling (risk of hidden periodicity bias) should be acknowledged.

Author's Response:

• Thanks, we incorporated it as follow in the limitation section

Lastly, there may be a potential of risk of hidden periodicity bias when systematic sampling applied.

Reviewer’s Q:

Data collection: It would help to specify whether inter-rater reliability was checked among data collectors when assessing phlebitis scores, to strengthen methodological rigor.

Author's Response:

• Be honest we did not conduct inter-rate reliability. This may enhance methodological rigor. What we did was when the data collector decided the score the supervisor cross check again and if there was discrepancy, it was solved with a discussion with other staffs who are working in the hospital based on the criteria.

Reviewer’s Q:

Data quality assurance: Translation and back-translation are appropriate, but a note on whether content validity or reliability testing of the questionnaire was performed would strengthen this section.

Author's Response:

• Thank you, we checked the reliability of the questionnaire and modification done the main document

Reviewer’s Q:

Data management and analysis: Testing of proportional hazards assumptions and checking for multicollinearity are strengths. However, the authors should mention how missing data (if any) were handled.

Author's Response:

• Thank you for your comment whereas, we were not faced this challenge. The nature of the data collection was not faced us for this challenge.

Reviewer’s Q:

Ethical consideration: The manuscript mentions that ethical approval was obtained, but the reference number or formal approval letter is not provided. For transparency and compliance with journal standards, it is recommended to include the ethics committee approval reference number and, if available, the date of approval. This ensures readers can verify that the study was conducted in accordance with ethical guidelines.

Author's Response:

• You are correct we did modification on it, thanks. Also, we attached the letter as well.

Ethical clearance was obtained from Debre Markos University College of Medicine and Health Sciences Research Ethics and Approval Committee, with reference number RCSTTD/395/01/17, dated 01/11/2024.

Reviewer’s Q:

Result

Socio-demographic characteristics: The reporting of contraceptive use could be clarified further; Is the denominator all female participants (168), or only non-pregnant ones?

Author's Response:

• It is only non-pregnant females, and we tried to correct in the main document as follow;

Additionally, 45(26.8%) of the non-pregnant female participants reported using contraceptives, with 35 (20.8%) specifically using Depo-Provera

Reviewer’s Q:

Clinical, Infusion, cannula-related characteristics: Nearly three-fourths of the study participants were admitted with a single diagnosis” – It would be clearer to provide the exact number and percentage instead of a vague phrase (“nearly three-fourths”).

Author's Response:

We incorporated the exact number with its percentage.

Reviewer’s Q:

The phrase “three-fourths of the participants had a normal BMI” is too approximate. Giving exact numbers and percentages would be more precise.

Author's Response:

We modified it as follow;

Regarding body mass index, 223(59.9%) of the participants had a normal body mass index, whereas 26 participants (7.0%) could not be categorized due to general body swelling, pregnancy, or disability

Reviewer’s Q:

Treatment Data (IV fluids, drugs, injections, blood transfusion): The flow is a bit disorganized. It mixes different interventions without grouping them logically. Consider restructuring: first fluids/drugs, then injection frequency, then transfusion.

Author's Response:

Thanks, we did correction as follow; Out of the total participants, 345 (92.7%) received intravenous drugs, and 192 participants (51.6%) received two or fewer injections per day. Whereas, 155 (41.7%) participants received intravenous fluids, and 37(10.0%) participants received blood transfusions.

Reviewer’s Q:

Almost half patients (51.61%) received two or fewer injections per daily” → needs correction for grammar: “…per day.

Author's Response:

• Thanks, we did correction.

Reviewer’s Q:

Again, percentages like 29.03%, 35.19%, 41.67%, etc. are overly precise. Rounding to one decimal place (e.g., 29.0%, 35.2%, 41.7%) is more standard in reporting.

Author's Response:

• Thanks, we did correction.

Reviewer’s Q:

Predictors of phlebitis: Use either “Cox proportional

---

## [Decision Letter · Decision Letter 1]

22 Oct 2025

Time to phlebitis onsite and its predictors among admitted patients with peripheral intravenous cannulas at Debre Markos Comprehensive Specialized Hospital, Northwest Ethiopia, 2024/2025: a prospective follow-up study.

PONE-D-25-39587R1

Dear Dr. Mitiku,

We’re pleased to inform you that your manuscript has been judged scientifically suitable for publication and will be formally accepted for publication once it meets all outstanding technical requirements.

Kind regards,

Federica Canzan

Academic Editor

PLOS ONE

Additional Editor Comments (optional):

Reviewers' comments:

Reviewer's Responses to Questions

**Comments to the Author**

Reviewer #1: All comments have been addressed

Reviewer #2: All comments have been addressed

2. Is the manuscript technically sound, and do the data support the conclusions?

Reviewer #1: Yes

Reviewer #2: Yes

3. Has the statistical analysis been performed appropriately and rigorously?

Reviewer #1: Yes

Reviewer #2: Yes

4. Have the authors made all data underlying the findings in their manuscript fully available?

Reviewer #1: Yes

Reviewer #2: Yes

5. Is the manuscript presented in an intelligible fashion and written in standard English?

Reviewer #1: Yes

Reviewer #2: Yes

Reviewer #1: I would like to thank the authors for their careful and thorough revisions. All my previous comments have been adequately addressed, and the manuscript has improved significantly.

Reviewer #2: (No Response)

**Do you want your identity to be public for this peer review?** For information about this choice, including consent withdrawal, please see our Privacy Policy

Reviewer #1: No

Reviewer #2: No

---

## [Editor Report · Acceptance letter]

PONE-D-25-39587R1

PLOS ONE

Dear Dr. Mitiku,

I'm pleased to inform you that your manuscript has been deemed suitable for publication in PLOS ONE. Congratulations! Your manuscript is now being handed over to our production team.

Kind regards,

on behalf of

Professor Federica Canzan

Academic Editor

PLOS ONE